# Trends of Dietary Intakes and Metabolic Diseases in Japanese Adults: Assessment of National Health Promotion Policy and National Health and Nutrition Survey 1995–2019

**DOI:** 10.3390/jcm11092350

**Published:** 2022-04-22

**Authors:** Muhammad Fauzi, Indri Kartiko-Sari, Hemant Poudyal

**Affiliations:** 1Department of Diabetes, Endocrinology, and Nutrition, Graduate School of Medicine, Kyoto University, Kyoto 606-8501, Japan; mfauzi@kuhp.kyoto-u.ac.jp; 2Asian Nutrition and Food Culture Research Center, Jumonji University, Saitama 352-8510, Japan; indri.kartiko@gmail.com; 3Population Health and Policy Research Unit, Graduate School of Medicine, Kyoto University, Kyoto 606-8501, Japan

**Keywords:** Health Japan 21, gender-based policy, dietary pattern

## Abstract

Health Japan 21 is Japan’s premier health promotion policy encompassing preventive community health measures for lifestyle-related diseases. In this repeated cross-sectional survey, we report 24-year trends of type 2 diabetes mellitus (T2DM), obesity, hypertension, and their association with dietary intakes to evaluate Health Japan 21’s impact and identify gaps for future policy implementation. We analyzed data from 217,519 and 232,821 adults participating in the physical examination and dietary intake assessment, respectively, of the National Health and Nutrition Survey 1995–2019. Average HbA1c and BMI have significantly increased along with the prevalence of T2DM and overweight/obesity among males. Despite a significant decrease in daily salt intake, the decline in the combined prevalence of Grades 1–3 hypertension was non-significant. Seafood and meat intakes showed strong opposing trends during the study period, indicating a dietary shift in the Japanese population. Neither salt nor vegetable/fruit intake reached the target set by Health Japan 21. Metabolic disease trend differences between males and females highlight the need for a gender-specific health promotion policy. Future Health Japan 21 implementation must also consider locally emerging dietary trends.

## 1. Introduction

Lifestyle-related diseases (LRDs) cause over 60% of deaths and collectively account for over a third of all healthcare expenditure in Japan [1]. The increases in the prevalence of LRDs further complicate the access to health in a rapidly aging population, most of whom live in single-occupancy households or are cared for by their elderly partners [2]. Among LRDs, diseases of the circulatory system such as stroke and ischemic heart diseases make up a significant portion of the health burden in Japan [1].

Hypertension, type 2 diabetes mellitus (T2DM), and obesity (hereinafter collectively referred to as metabolic diseases) are major risk factors of LRDs with well-established nutritional etiology, particularly excessive salt and energy intakes, as well as physical inactivity. Given the historically high salt intake in Japan, several policies have been implemented that have helped reduce the average salt intake by ~4 g/day [3,4]. Similarly, total energy intake showed a decreasing trend between 1995 and 2016 in both genders, with lower energy intake (%) from protein but greater energy intake (%) from fat [4]. Yet, the prevalence of overweight, obesity, and T2DM has been increasing, particularly among males [5,6]. Moreover, about 31 million out of the estimated 43 million hypertensive adults have poorly controlled hypertension [7].

Several preventive community public health measures focusing on risk factors of LRD and elderly care have been implemented to promote healthy aging in Japan [8,9,10,11]. Health Japan 21 is the current health promotion plan in Japan that stipulates the policies, ideas, and specific target goals for health promotion, covering the entire lifecycle from fetal to advanced age. In its first term (2000–2012), the policy outlined 79 targets based on the data from the annual National Health and Nutrition Survey (NHNS) [12] under nine focus areas: nutrition and diet, physical activity and exercise, rest and promotion of mental health, tobacco smoking, alcohol intake, and dental health [13]. Subsequently, it was revised and relaunched as Health Japan 21 Second Term (2013–2022) with fifty-three targets and five pillars: life expectancy extension and social disparities reduction, LRDs prevention, social life engagement improvement, social environment establishment, and lifestyle improvement, which includes nutrition and dietary habits, physical activity and exercise, rest, drinking alcohol, smoking tobacco, and oral health [14].

In this study, we report the 24-year trend of T2DM, overweight/obesity, and hypertension in Japan from the NHNS (1995–2019) and the consumption of macronutrients and food groups in the Japanese diet in relation to the targets set under Health Japan 21 to identify gaps for future health promotion policy implementation in Japan.

## 2. Materials and Methods

### 2.1. Study Design and Data Retrieval

The prevalence of T2DM, overweight/obesity, hypertension, and Japanese adults’ dietary intake data (≥20 years old) from 1995 to 2019 were retrieved from NHNS. In brief, the Ministry of Health, Labor, and Welfare (MHLW) of Japan has been supervising an annual National Nutrition Survey since 1945, later renamed NHNS in 2002. The survey is conducted in November each year, except in 2012, when it was conducted from late October to early December. NHNS is a repeated cross-sectional survey with different participants drawn each year. In each survey, all prefectures were represented except in 2011 and 2016, when certain regions were excluded due to natural disasters. In the 2011 survey, Iwate, Miyagi, and Fukushima prefectures were excluded due to the Great East Japan Earthquake. Kumamoto prefecture was excluded in 2016 due to the Kumamoto earthquakes.

Physical examination and blood biochemistry were reported in each survey with slightly different content, as reported elsewhere [15]. The survey on T2DM in Japan was first conducted under the National Diabetes Surveys in 1997 and 2002 and subsequently integrated into NHNS. HbA1c was measured using the Japan Diabetes Society (JDS) standard until 2011 and under the National Glycohemoglobin Standardization Program (NGSP) standard since 2012. Data prior to 2012 have been converted to the NGSP standard using the formula: NGSP (%) = 1.02 × JDS (%) + 0.25 [16].

NHNS defines people with HbA1c ≥ 6.5% (NGSP), with or without insulin treatment or oral hypoglycemic medication, as “people strongly suspected of having diabetes.” In this study, we referred to this population as people with T2DM. Height and body weight were measured with the detailed procedure outlined elsewhere [17]. Those with a body mass index (BMI, kg/m^2^) of 25 ≤ BMI < 30 were classified as overweight and ≥30 as obese. Blood pressure was measured using the Riva-Rocci mercurial sphygmomanometer and JIS manchette, except in 2019, when a hybrid sphygmomanometer was used. Hypertension classifications were divided into three groups based on systolic blood pressure (SBP mm/Hg) and/or diastolic blood pressure (DBP mm/Hg), described by The Japanese Society of Hypertension as follows: Grade I (SBP 140–159 and/or DBP 90–99), Grade II (SBP 160–179 and/or DBP 100–109), and Grade III (SBP ≧ 180 and/or DBP ≧ 110). In this study, we included data from patients with treated and untreated hypertension.

Dietary assessment and identification have been explained in detail elsewhere [17]. Briefly, the dietary record that included data on all household members was given to the primary record keeper of the household by a registered dietitian. Before the implementation, written and verbal instructions on maintaining the record were provided to the participating households. The record keeper was requested to weigh ingredients used for food preparations or to approximate the proportion of food taken and leftovers when weighing was not possible. The recording day was freely chosen by the record keeper, excluding Sundays, national holidays, or ceremonial days where special meals were consumed. The dietary record was then collected, confirmed, and corrected (if necessary) by a registered dietitian shortly after recording.

MHLW annually publishes the survey results, while the targets of Health Japan 21 are published by the National Institute of Health and Nutrition. In this study, we analyzed physical examination and dietary records of participants aged ≥20 years, males and females, excluding lactating or pregnant women. The number of participants and participant characteristics of each survey are summarized in Appendix A.

### 2.2. Statistical Analysis

In descriptive statistics, continuous variables were expressed as the mean ± SD, and categorical variables were expressed as the number and percentage (%). For trend analyses, we adopted general linear regression to determine the degrees of change in metabolic diseases using diagnostic measurements and the intakes of macronutrients and food groups. The analysis parameter was set to account for N and scatter among replicates from the standard deviation. No adjustments were made for multiple comparisons. All statistical calculations were performed with GraphPad Prism 9.0 (GraphPad Software, San Diego, CA, USA). The statistical significance threshold was set at *p* < 0.05.

## 3. Results

### 3.1. Prevalence of T2DM, Overweight/Obesity, and Hypertension

The national prevalence of T2DM doubled from 9.8% in 1997 to 19.7% in 2019 among Japanese males, while a modest but significant increase from 7.1% in 1997 to 10.8% in 2019 was observed in females (Figure 1A). HbA1c levels increased substantially in both genders between 1997 and 2002, although the effect was stronger among males than in females (Figure 1B). The combined prevalence of overweight and obesity increased from 23.8% to 33.0% among males and a non-significant increase from 20.8% to 22.3% among females in the past 24 years (Figure 1C). The prevalence of overweight and obesity is separately reported in Appendix A. The increase in BMI was also more apparent in males than in females (Figure 1D). The average BMI of males increased from 23 kg/m^2^ in 1995 to 23.9 kg/m^2^ in 2019, while the change in BMI in females only amounted to 0.03 kg/m^2^ during the study period, despite decreasing trends during 2006–2016.

Average SBP (Figure 2A) and DBP (Figure 2B) significantly declined among males and females during 1995–2019. This decline was reflected in Grade 2 and Grade 3 but not in Grade 1 or the combined prevalence of hypertension (data not shown) in males and females. In males, Grade 1 hypertension prevalence increased from 28.2% in 1995 to 32.3% in 2018 (Figure 2C). However, it drastically decreased to 24.8% in 2019 with the change in measurement method from the Riva-Rocci mercurial sphygmomanometer and JIS manchette to the hybrid sphygmomanometer (Figure 2C). Meanwhile, Grade 2 among males decreased from 10.4% to 7.3%, and Grade 3 decreased from 2.9% to 1.4% during the study period (Figure 2C). Grade 1 hypertension slightly decreased from 22.8% to 20.9% in females, while Grade 2 decreased from 7.9% to 5.0%, and Grade 3 decreased from 2% to 0.7% (Figure 2D).

No targets were explicitly set for the prevalence of T2DM or HbA1c in Health Japan 21. It does, however, set the target to reduce the percentage of individuals with HbA1c (NGSP) ≧ 8.4% to under 1%. For the combined prevalence of overweight and obesity, the target was set to limit the prevalence to under 28% among males and 19% among females by 2022. However, the combined prevalence of overweight and obesity of 33.0% in males and 22.3% in females exceeded the target in 2019. The target for average SBP was 134 mmHg (3-point reduction from 1995) for males and 129 mmHg (1.6-point reduction from 1995) for females by 2022. The average SBP target was achieved in females in 2018 and males in 2019.

### 3.2. Consumption of Macronutrients

Figure 3 describes the intake of macronutrients from 1995 to 2019. Total energy intake significantly decreased from 2327 to 2141 kcal in males and 1855 to 1717 kcal in females (Figure 3A). The intake of total carbohydrate and sugar sweeteners significantly decreased during this period, while fiber intake showed an upward trend in 2019, with overall significance in males but not females (Figure 3A–D) [it should be noted that NHNS adopted the American Association of Analytical Chemists (AOAC) 2011.25 analytical method to estimate dietary fiber intake in 2019]. The trend for total fat intake between 1995 and 2019 was not significantly different in males or females (Figure 3E). However, the intake of animal and plant fat showed statistically significant trends in both genders (Figure 3F,G). In contrast, the total, animal, or plant protein intakes have significantly declined in both genders (Figure 3H–J). The change in the protein/fat/carbohydrate energy intake ratio is summarized in Appendix A.

### 3.3. Consumption of Food Groups and Salt Intake

There has been a substantial increase in meat intake (Figure 4A) since 2001, while seafood intake (Figure 4B) has correspondingly declined in both genders. Vegetable intake significantly changed in both groups (Figure 4C), while the total population intake in 2019 (280 g/day) was still far below the 2022 target (350 g/day). Fruit was the only food group that females consumed more than males (Figure 4D). However, both genders showed significantly decreased daily fruit intake over the study period (Figure 4D). The intake of grains has also declined in both genders (Figure 4E), in line with the decline in total carbohydrate intake. Finally, we observed significant declines in salt consumption from 15 to 10.9 g in males and 13 to 9.3 g in females (Figure 4F). However, salt intake remains substantially higher than the Health Japan 21 target of 8 g/day.

## 4. Discussion

The present study investigated the changes in metabolic diseases and dietary intakes in Japanese adults over two decades. Our results showed that T2DM, overweight, and obesity have increased in the Japanese population, especially among males. Meanwhile, the prevalence of Grade 2 and 3 hypertension has declined, but the prevalence of Grade 1 or the combined prevalence of hypertension has remained largely unchanged. Furthermore, in terms of macronutrient intakes, carbohydrate and protein intakes have declined during the study period, while the intake of fat, especially plant fat, has increased. Similarly, the intake of several food groups, including grains, vegetables, fruits, and seafood, continues to decline, with a corresponding increase in meat intake. We also observed that several survey items, such as the prevalence of overweight/obesity, national average HbA1c, and vegetable/fruit intake, were far from the Health Japan 21 targets for 2022. Meanwhile, the average daily salt intake targets may be achieved but will need aggressive policy implementation.

The trends for the prevalence of metabolic diseases differed by gender. Our results showed an increasing national average of HbA1c from 1997 to 2019 in both sexes, with the male group showing a more significant escalation. In line with HbA1c data, we found a greater increase in T2DM prevalence among males than females. Interestingly, a similar trend was also observed in BMI and the prevalence of overweight and obesity, where males showed a steady increase while the female group tended to be stable over the years. We also noted that the decrease in salt intake and improvement of SBP and DBP were independent of the prevalence of Grade 1 hypertension in both sexes.

Although our analysis confirms the previously reported decline in energy consumption in the Japanese population [4], the steady increase in the prevalence of T2DM and overweight/obesity suggests that energy intake may not be the sole determinant of metabolic diseases in Japan, and changes in food group consumption may account for some of the observed trends. For instance, a large Japanese cohort study recently showed an inverse relationship between vegetable intake and body weight change but reported a positive relationship between fruit intake and body weight [18]. Furthermore, large observational studies suggest that high animal protein intake is associated with a greater risk of T2DM, cardiovascular, and all-cause mortality than plant protein [19,20]. However, plant protein’s advantage over animal protein has not been observed in randomized controlled trials recruiting patients with T2DM, obesity, or both, and a higher protein intake, regardless of the source, may have potential benefits in glycemic, adiposity, and blood pressure control [21,22,23]. We observed a marked decline in protein intake from plant and animal sources in the Japanese population.

Additionally, observational studies have shown an inverse relationship between vegetable fat intake and the incidence of T2DM [24]. Although we observed a marked increase in total fat intake in Japan, mostly due to increased plant fat intake, the prevalence of T2DM and the national average HbA1c continue to increase. A recent analysis of two Japanese nationwide diabetes registries noted an increase in fat and meat intake with significant decreases in the intake of fish, soy products, fruits, and vegetables among people with diabetes over a 20-year period [25]. Although Japan is recognized for its seafood diet, our findings concurringly showed that the Japanese population consumed less seafood and more meat in the past two decades. According to the Food and Agricultural Organization, the Japanese population consumed 67.3 kg of seafood annually in 2000 while consuming 47.6 kg/year in 2013 [26]. In this study, we observed a significant downward trend in seafood intake in both males and females. Fish intake has been associated with lower risks of T2DM, obesity, and hypertension [27,28]. These beneficial effects of fish have been attributed to enhancing glucose-stimulated insulin secretion, protecting β-cells, and increasing insulin sensitivity by n-3 PUFA [29]. Furthermore, n-3 PUFA may promote lipid oxidation and enhance energy utilization by altering several key protein expressions and suppressing the renin-angiotensin-aldosterone system to produce a positive outcome in obesity and hypertension [30,31].

In contrast, meat intake showed a significant increasing trend during the study period. Meat is a key source of protein and fat, and its effects on health have attracted much attention regarding its association with cardiovascular diseases and diabetes. Associations between meat consumption and T2DM have been reported in various populations [32]. The adverse effect of meat consumption on T2DM parameters, such as insulin and HbA1c, may be mediated by increasing visceral adiposity [33]. In addition, some studies have highlighted that processed meat, but not unprocessed meat, was associated with T2DM, emphasizing that preceding preparation may be an important determinant of its effect on metabolic wellbeing [34].

In any case, the shift from seafood to meat as the primary dietary protein source is consistent with the continuous westernizing of the Japanese diet [17,35]. For almost half of the century until 2005, food balance sheet data showed an increased per-capita supply of meat/poultry and fats/oil, a decrease in rice supply, and a relatively stable supply of seafood, fruits, and vegetables [36]. These trends indicate a shift from the traditional Japanese diets based on white rice, fish, soybean products, seaweeds, and green tea [37]. The adverse effect of the westernized diet is apparent in the Japanese population who reside in Japan and in Japanese expatriates who must adapt their diet to the local diet [38,39].

A rapidly westernizing dietary pattern and the traditionally high salt intake pose a significant challenge for health promotion in Japan beyond the well-established relationship between salt intake and the prevalence of hypertension. High salt preference was positively associated with undiagnosed diabetes in men, mediated by BMI and central obesity [40]. On the contrary, a population study found that salt intake (about 4.3 g/day) was higher among women with newly diagnosed diabetes [41]. Furthermore, a study investigating the degree of salt intake’s effect on insulin resistance in hypertension-prone subjects suggested an interaction between salt intake, the renin-angiotensin-aldosterone system, and insulin action in men. On the contrary, salt sensitivity was directly related to body weight gain in women, which suggested a mechanism difference in salt sensitivity between both sexes despite the similar age and BMI [42]. Meanwhile, the direct positive relationship between salt intake and hypertension has been extensively reviewed [43]. However, the decreasing salt intake did not correspond with the overall prevalence of hypertension in the Japanese population. Although we observed a decline in the prevalence of Grade 2 and Grade 3 hypertension, this was offset by Grade 1 hypertension prevalence, which is predominant in the Japanese population. In females, both Grade 2 and Grade 3 hypertension showed improvement, while Grade 1 tended to be stable. The aggressive treatment and control of hypertension in Japan [44] and other lifestyle factors such as a reduction in smoking [45] may have contributed to the observed trends in hypertension prevalence.

Health Japan 21 set targets with goals to be reached by 2022 to track the progress of health promotion in reducing metabolic disease trends since it was first started in 2000. However, we found several challenges which need to be resolved to achieve the target. First, T2DM prevalence cannot be directly compared to the target since it is expressed in absolute projection numbers of 10 million patients with diabetes by 2022, hindering evaluation attempts by external investigators. Second, the prevalence of overweight/obesity in both genders has already surpassed the target set under Health Japan 21 in 2019, and with another three years of potentially worsening trends before the 2022 survey data, it is likely to pose a significant challenge for future policy implementation. Third, the lack of improvement in Grade 1 hypertension and its contradiction with decreasing salt intake and blood pressure sparks the question of whether solely focusing on salt intake is enough to prevent mild hypertension. Fourth, the current health promotion campaign excessively highlights salt reduction rather than balanced nutrition intake from food groups. The findings of this study suggested that less meat and more seafood intake, for instance, are expected to improve favorable metabolic disease trends. 

Fifth, starkly different trends among males and females, both in the prevalence of disease and dietary intake, indicate the need for a gender-based health policy. Our observation concurs with an earlier prospective cohort study which reported worst cardiometabolic profile among males compared to females irrespective of age group [45]. While biological factors such as the propensity to accumulate abdominal fat among males, which is strongly associated with metabolic diseases [46,47], account for some of these gender-based differences, several unhealthy lifestyle factors such as smoking, excessive alcohol intake, fast eating, and late-night meals are also more prevalent among Japanese males [45]. Moreover, a positive association between fish intake and lipid metabolism have been reported among males but not in females, indicating that the decline in seafood consumption in the Japanese diet may be more consequential for males than females [48]. Further, our results showed that females consumed more fruits compared to males. While we refrain from drawing conclusions from the differences in fruit intake alone, it reflects different lifestyles between genders [49]. In any case, data on gender differences in health behaviors in the Japanese population are scarce, and research in this domain must be prioritized to guide future gender-based policy.

Finally, future health promotion goals must consider the rapid westernization of the Japanese diet along with increasing sedentary behavior. Several studies have highlighted the decline in step-defined physical activity in Japanese adults, especially among young and middle-aged adults, and its association with overweight/obesity, blood pressure, HbA1c, and cardiovascular and all-cause mortality [50,51,52,53,54]. Likely, the increasingly sedentary lifestyle in Japan is further exacerbating the effects of the rapidly changing Japanese diet.

## 5. Conclusions

In conclusion, our findings highlight the need to emphasize gender-specific health promotion policy and balanced nutrition in line with the emerging dietary trends. The provisions under “Promotion of health in communities” of Health Japan 21 may provide the scope for formulating specific plans for the promotion of health in the manner best suited to the actual situation of that particular area (local plans) by enlisting the cooperation of residents and various community health organizations.

## Figures and Tables

**Figure 1 jcm-11-02350-f001:**
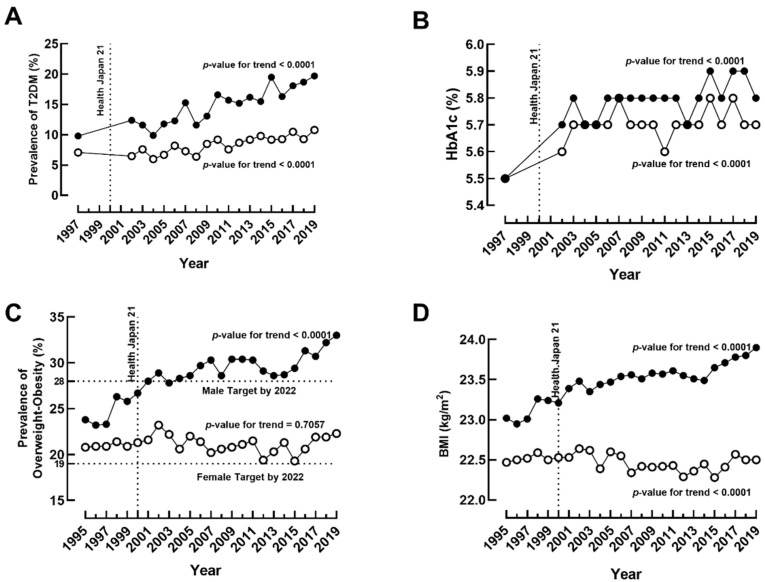
The prevalence of metabolic diseases and their predictors: (**A**) the prevalence of type 2 diabetes mellitus (T2DM) from 1997 to 2019, (**B**) the average annual glycated hemoglobin (HbA1c) from 1997 to 2019, (**C**) the prevalence of overweight/obesity from 1995 to 2019, and (**D**) the average annual body mass index (BMI) from 1995 to 2019 among males (●) and females (○).

**Figure 2 jcm-11-02350-f002:**
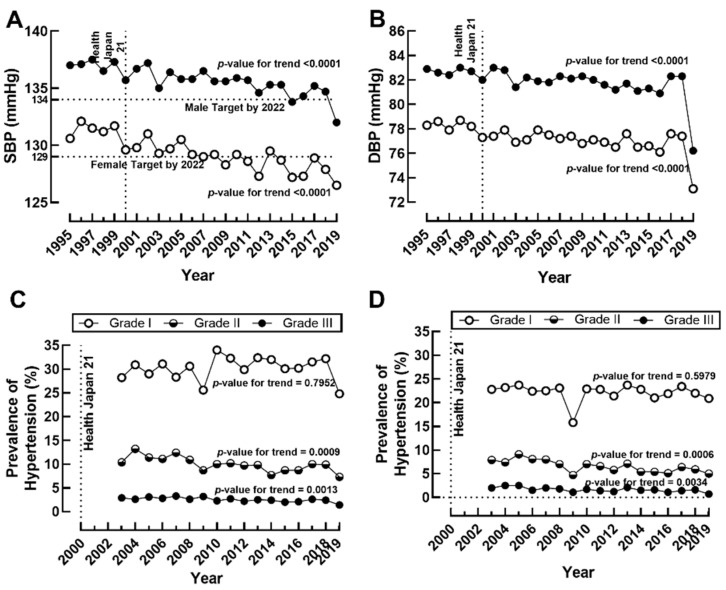
The annual average blood pressure and prevalence of hypertension: (**A**) Mean annual national systolic blood pressure (SBP) and (**B**) mean annual national diastolic blood pressure (DBP) from 1995 to 2019 among males (●) and females (○). The prevalence of hypertension by grade among (**C**) males and (**D**) females from 2003 to 2019.

**Figure 3 jcm-11-02350-f003:**
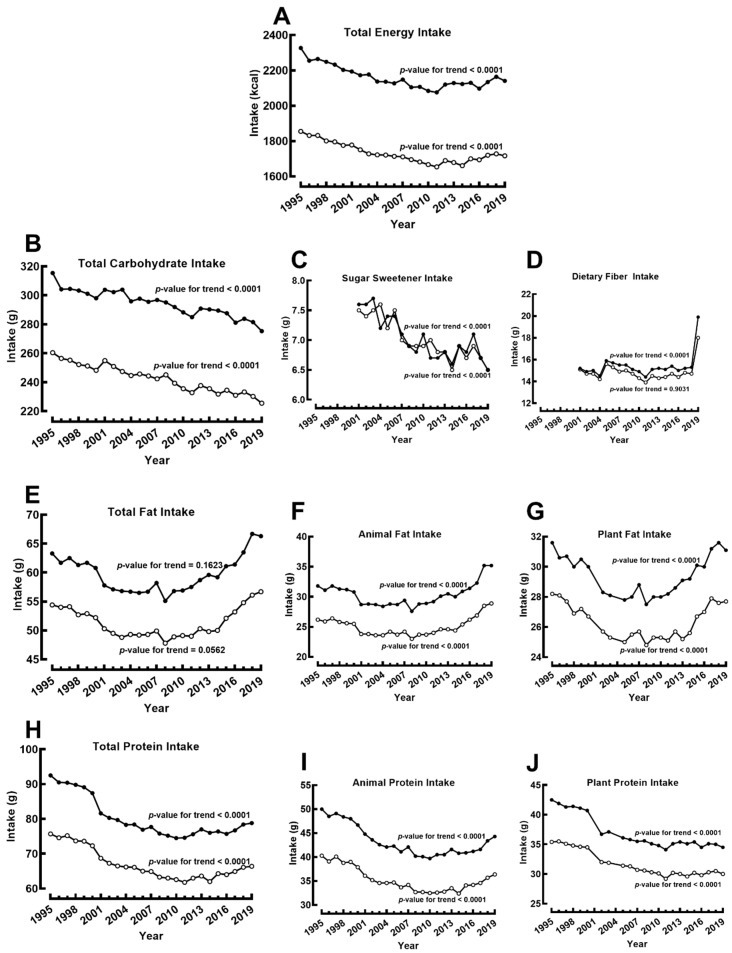
Changes in macronutrient intake from 1995 to 2019 among males (●) and females (○). (**A**) Total energy intake, (**B**) total carbohydrate intake, (**C**) intake of sugar sweeteners, (**D**) intake of dietary fibers, (**E**) total fat intake, (**F**) animal fat intake, (**G**) plant fat intake, (**H**) total protein intake, (**I**) animal protein intake, and (**J**) plant protein intake.

**Figure 4 jcm-11-02350-f004:**
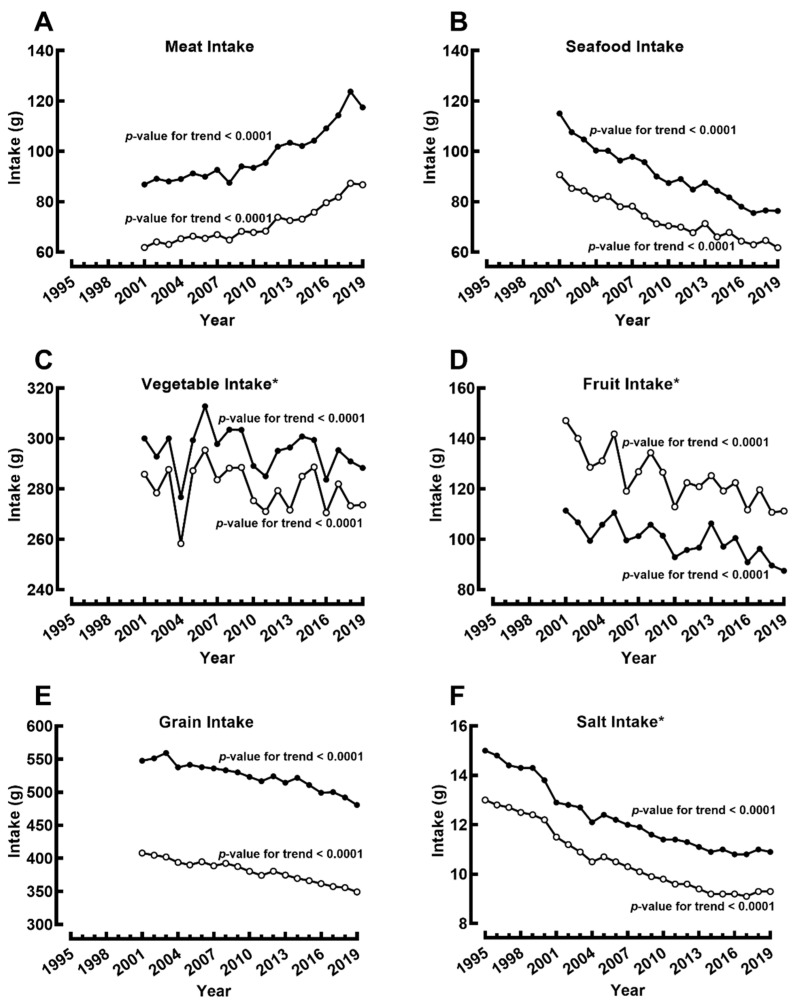
Changes in food group and salt intakes from 1995 to 2019 among males (●) and females (○). (**A**) Total meat intake, (**B**) total seafood intake, (**C**) total vegetable intake, (**D**) total fruit intake, (**E**) total grain intake, and (**F**) total salt intake. * The 2022 targets of these parameters are set under Health Japan 21 as follows: Mean 350 g/day intake of vegetables, 70% of individuals consuming more than 100 g/day of fruit, and decrease the mean salt intake to 8 g/day.

## Data Availability

The data presented in this study are openly available from https://www.mhlw.go.jp/bunya/kenkou/kenkou_eiyou_chousa.html and https://www.nibiohn.go.jp/eiken/kenkounippon21/en/eiyouchousa/keinen_henka_eiyou_select.html (accessed on 31 March 2022).

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
