# Peer review of "Trends of Dietary Intakes and Metabolic Diseases in Japanese Adults: Assessment of National Health Promotion Policy and National Health and Nutrition Survey 1995–2019"

_jcm, 2022, doi:10.3390/jcm11092350_

Round 1
Reviewer 1 Report
The authors have addressed and revised their manuscript according to my former comments. This article is now more concise and should be of interest to the readers of JCM.
Author Response
We thank the reviewer for their suggestions and comments which have helped improve the overall quality of the manuscript.
Reviewer 2 Report
There a few comments:
- I'm wondering whether the data is a bit old, as the latest survey year is 2018. Especially experienced the COVID 19, the trend of cardiometabolic diseases and nutrition intake would make a huge difference
- It is not very clear how you run the analyses. You used GraphPad Prism 9.0, which seems not suitable for complex statistic analyses. It would be better to provide detailed analysis methods, including what modelling being used, what confounders being controlled etc.
- As per discussed that meat consumption can have a different impact on the cardiometabolic outcomes, such as processed meat vs unprocessed meat or red meat. Do you have these sub groups of meat consumption, instead of a generic meat consumption to be able to identify the difference of meat consumption?
- Give the detected gender difference, what would be informed in the future policies or intervention programs? What are the potential drivers for the difference?
Author Response
We thank the reviewer for their additional suggestions and comments which have helped improve the overall quality of the manuscript. We have included the point-by-point response to the comments below:
1. I'm wondering whether the data is a bit old, as the latest survey year is 2018. Especially experienced the COVID 19, the trend of cardiometabolic diseases and nutrition intake would make a huge difference.
RESPONSE: We have now included the 2019 data set and updated the manuscript accordingly with some minor changes in the results. However, the release of 2020-21 data by the Japanese Ministry of Health Labour and Welfare has been delayed due to the pandemic. Nevertheless, we agree that it will be interesting to look at dietary changes due to the pandemic and will be including this assessment in a future manuscript.
2. It is not very clear how you run the analyses. You used GraphPad Prism 9.0, which seems not suitable for complex statistic analyses. It would be better to provide detailed analysis methods, including what modelling being used, what confounders being controlled etc.
RESPONSE: We have now added more details to the statistical analysis. We only report weighted but unadjusted data and analyzed trends using simple linear regression in this paper and are currently in the process of conducting a more extensive multivariate analysis (for which Prism is undoubtedly underpowered).
3. As per discussed that meat consumption can have a different impact on the cardiometabolic outcomes, such as processed meat vs unprocessed meat or red meat. Do you have these sub groups of meat consumption, instead of a generic meat consumption to be able to identify the difference of meat consumption?
RESPONSE: While this is of considerable interest, unfortunately, the NHNS does not report sub-groups of meat intake by preparation method (processed vs. unprocessed).
4. Give the detected gender difference, what would be informed in the future policies or intervention programs? What are the potential drivers for the difference?
RESPONSE: We have now expanded the discussion on the potential drivers and implications of gender differences on pages 9 and 10 (lines 489-513).